# DATA AUGMENTATION VIA GENOMIC FOUNDATION MODELS FOR PSEUDOKNOT-INCLUSIVE RNA SECONDARY STRUCTURE PREDICTION

## ABSTRACT

Rapid advancements in genomic foundation models (GFMs) have delivered a series of breakthroughs across a diverse set of tasks for RNA, however RNA Secondary Structure Prediction (SSP) remains a pivotal task in computational biology. Despite achieving breakthroughs in pseudoknot-free SSP, where state-of-the-art models can achieve above 80% macro-F1, performance on the pseudoknot-inclusive problem remains stagnate, with previous methods achieving below 50% macro-F1 on all three of our test-sets. This is due to a variety of challenges: a ginormous search space that limits heuristic performance, the major class imbalance problem that limits the usual classification methods, and the inherent lack of data that limits deep learning methods. Further data acquisition is implausible due to requiring extensive biological resources and being associated with a high cost. In this work, we propose a novel approach to enhance RNA secondary structure prediction by implementing a novel data augmentation technique, specifically designed for the pseudoknot-inclusive SSP problem. Our method leverages masked language modelling (MLM) with a surrogate model to produce accurate and useful data augmentations, and we further utilise uncertainty quantification strategies to identify areas within the dataset where augmentation is most effective - thereby helping to mitigate the class imbalance problem, and further improving on the generalisability of the models. We further extend three GFMs, and fine-tune them using the augmented datasets to demonstrate the efficacy and high performance of the models. Notably, the newly extended and augmented models achieve state-of-the-art performance, achieving over 89% F1 on RNAStrAlign, and over 66% F1 on bpRNA test sets respectively. We therefore highlight the effectiveness of data augmentation for genomic data, and release our code and datasets to assist future researchers.

## 1 INTRODUCTION

Ribonucleic Acid (RNA) is a crucial biological molecule that is essential for the function and regulation of living organisms (Morris & Mattick, 2014). Recently, it has garnered significant interest due to its potential applications in therapeutics and vaccine development (Wang & Farhana, 2024; Zhu et al., 2022). RNA sequences are composed of the nucleotides Adenine (A), Cytosine (C), Guanine (G), and Uracil (U), which form bonds, and fold into secondary structures. The arrangement of nucleotides is critical for the formation of these structures, as it heavily influences the folding process, which enables RNA to interact with various cellular components (Ganser et al., 2019). This folding process relies on base-pair interactions, with both the length of the nucleotide sequence and the specific types of interactions being important for the stability and functionality of the folded RNA (Sanchez de Groot et al., 2019). The secondary structure serves as a foundation for the creation of local structures and motifs within the RNA (Tinoco & Bustamante, 1999).

The sequence of nucleotides that RNA is made up of can be seen as it's own unique language, with an inherent structure. This is the essence of Genomic Foundation Models (GFMs), where each nucleotide is fed into a large scale model, which can understand and model the underlying principles of biology (Nguyen et al., 2024a). This central idea behind GFMs has sparked an interest in applying large-scale models to genomic problems, specifically in RNA, thus, there are many previous works,

including several benchmarks that include secondary structure prediction (SSP) algorithms (Runge et al., 2024; Ren et al., 2024). Yet these benchmarks do not consider the pseudoknot, a key part of an RNA structure proven to have a profound impact on an RNA's functionality and efficiency (Giedroc et al., 2000). A pseudoknot is where base-pairing occurs in a non-sequential manner, thereby causing the RNA to fold into itself, thus creating complex knot-like motifs within the secondary structure. Despite its importance, GFMs only consider the pseudoknot-free SSP problem (Yang & Li, 2024; Fu et al., 2021; Wayment-Steele et al., 2022; Sato et al., 2021), and have yet to be extended to the pseudoknot-inclusive prediction problem. These works cite three key explanations for this; the lack of data (Wayment-Steele et al., 2022), the huge search-space of the problem (Janssen & Giegerich, 2014), and the extreme class imbalance (Reidys et al., 2011).

The lack of data is of paramount importance, however data acquisition is resource-intensive, time-consuming and expensive, especially for pseudoknot-inclusive data (Vicens & Kieft, 2022). As we aim to utilise large-scale GFMs, we must consider the quantity and quality of the data, which will directly impact model performance and task generalisation (Ramponi & Plank, 2020) Furthermore, the computational complexity required for secondary structure prediction with pseudoknots has been proven to be NP-Hard (Akutsu, 2000; Lyngsø & Pedersen, 2000), thus dynamic programming-based methods require polynomial time ($O(n^4) - O(n^6)$) where $n$ represents the length of the sequence. Lastly, regarding the class imbalance challenge, pseudoknot-free secondary structures consist of base pairs and unpaired nucleotides, represented by "(", ")", and ".". These three elements are relatively balanced in terms of their occurrence within an RNA secondary structure, however, the introduction of pseudoknots adds a range of non-canonical base pairs and long-range interactions, represented by "[", "]", "{", "}", "<", and ">". Due to the rarity of pseudoknots, a pronounced class imbalance arises within the dataset, making it challenging for models to generalise effectively (Reidys et al., 2011). This is illustrated in fig. 1, which demonstrates the severity of the class imbalance.

To address the lack of data, researchers have increasingly turned to data augmentation (Runge et al., 2019; Marouf et al., 2020), however, the usage of data augmentation for key application-based tasks, specifically in genomic-based tasks, remains under-utilised. In this work, we introduce a novel data augmentation method, utilising large-scale GFMs and uncertainty quantification to synthetically expand our data. A further advantage of GFMs is the large parameter-scales that can accurately model the complex relationship between RNA sequence and secondary structures, as to circumnavigate the large search-space. We further navigate the lack of data by providing two clear training and validation datasets, and by implementing a novel data augmentation methodology, tailor-

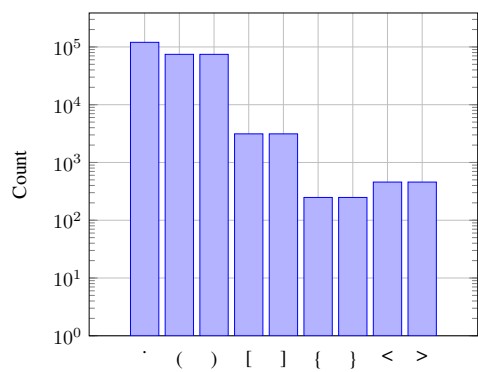

Figure 1: Dot-Bracket notation symbol count in bpRNA and RNAStrAlign datasets

made for the pseudoknot-inclusive SSP problem. Our data augmentation methodology is further employed to reduce the class imbalance on the training dataset, and thereby assist model generalisation.

We outline our contributions as the following:

- We introduce two key pseudoknot-inclusive datasets for training, validating and testing algorithms.
- We propose a novel data augmentation strategy, customised for the pseudoknot-inclusive SSP problem, to augment pseudoknot-inclusive training data.
- We demonstrate the efficacy of our method by extending three foundation models (OmniGenome, HyenaDNA, SpliceBERT) to the pseudoknot-inclusive SSP problem, showing significant improvement over previous methods.

## 2 RELATED WORK

The rapid development of Natural Language Processing (NLP) in recent years is underpinned by the demand for high-quality data (Feng et al., 2021). Obtaining high-quality data remains a significant challenge, particularly for genomic data, where data acquisition requires substantial time and resources to ensure biological validity (Vicens & Kieft, 2022). This lack of data is a fundamental problem across various areas of genomics, such as transcriptomics (Lacan et al., 2023), DNA sequences (Lee et al., 2023), and protein genotypes (Minot & Reddy, 2022). To address this problem, various methods have been employed, such as sequence-based augmentation (Lee et al., 2023), and the usage of generative models (Anand & Huang, 2018). However, there remains little usage of data augmentation for RNA-based tasks. The highly structured nature of genomic sequences prevent traditional data augmentation methods from being effective (Ching et al., 2018), as a simple change in the sequence could have a dramatic effect on the resulting structure, thus only context-aware data augmentation strategies are viable for this problem (Yang et al., 2018). Moreover, for GFMs, this necessitates the usage of Single Nucleotide Tokenisation (SNT), as to prevent misalignment between tokenised inputs and labels within text-based models (Yang & Li, 2024). As commonly used NLP data augmentation strategies are designed for unstructured text (Wei & Zou, 2019), they are unsuitable for application to highly-structured genomic data.

One method of mitigating the structured nature of genomic data is the usage of masked-language modeling (MLM), which has been recently utilised by researchers within the GFM space (Karollus et al., 2024; Hwang et al., 2024). However, this introduces its own set of challenges, such as the reliability of model predictions, which must be considered. One method of mitigating this challenge that has recently been explored is the usage of uncertainty quantification (UQ) within pre-trained language models (Wang et al., 2022; Liu et al., 2024). Uncertainty quantification has emerged as a vital technique for assessing and improving the reliability of model predictions, with various works (Kong et al., 2020; Wang et al., 2022) demonstrating an over-fitting problem in fine-tuned pre-trained models. This is due to the extremely large parameter size of the models, and the small amount of data being provided for fine-tuning model predictions (Kong et al., 2020). Apart from simply adding more data in the fine-tuning process, we further explore the effect that uncertainty quantification methods have on improving model performance.

## 3 METHODOLOGY

GFMs have demonstrated effectiveness in genomic context-based learning, accurately predicting masked nucleotide bases within sequences (Yang & Li, 2024). Given the highly context-dependent nature of RNA secondary structures, we propose to directly mask a portion of the original dataset and employ MLM to generate augmentations of experimentally observed sequences, thereby increasing the sample size per instance. Although MLM for data augmentation has been previously explored (Zhou et al., 2022; Li et al., 2022), we introduce a novel variation by incorporating uncertainty quantification into the methodology, designed to identify genomic regions with high or low uncertainty. Furthermore, we tailor this method specifically for the pseudoknot-inclusive secondary structure prediction (SSP) problem by leveraging known pseudoknots and prior data to enhance the effectiveness of the masking approach. We term this method *Sequence Location Masking* (SLM), which is designed to be seamlessly integrated into any GFM training pipeline by simply substituting the original training set with the augmented one.

The augmentation pipeline consists of several key steps. First, a GFM capable of MLM is selected for data augmentation. Second, the original dataset is loaded, ensuring it has been appropriately pre-processed. Third, an uncertainty quantification method is chosen; in accordance with current literature, we offer several approaches, including Softmax entropy and temperature scaling (Guo et al., 2017). Next, the proportion of nucleotide bases to be masked within each sequence is determined. This proportion must be carefully balanced, as insufficient masking may result in sequences that are too similar, potentially leading to overfitting during model training. Finally, the number of augmentations per genome instance is selected.

## 3.1 SEQUENCE LOCATION MASKING

To implement Sequence Location Masking (SLM), it is essential to first apply uncertainty quantification to identify regions within the sequence characterised by high uncertainty. This process involves several steps:

- **Secondary Structure Prediction**: The augmentation begins by inputting the sequence into a model to predict its secondary structure. This model provides a set of predictions that serve as the basis for subsequent uncertainty quantification.
- **Mask Location**:
  After predicting the secondary structure, uncertainty quantification is performed on the predictions to identify locations with high or low levels of uncertainty. Based on these uncertainty levels, regions with the highest uncertainty are selected for masking.
- **Augmentation Process**:
  The augmentation process involves calculating an uncertainty score for each possible nucleotide at every position in the sequence. For each position, the nucleotide associated with the lowest uncertainty score is selected as the predicted base, unless the uncertainty score exceeds a pre-specified cut-off value; in that case, the true label is used instead.
- **Incorporation of Uncertainty Cut-off**:
  A hyper-parameter $y$ is introduced, serving as an uncertainty cut-off value. This cut-off determines the threshold at which the true label (i.e., the actual nucleotide) is used instead of the predicted nucleotide. Specifically, if the uncertainty score at a given position exceeds the cut-off value $y$, the true nucleotide is retained. This process is repeated across all positions within the sequence.

We further describe this approach from an algorithmic standpoint, from the perspective of a single sequence for simplicity. We break this down into two key algorithms:

### 3.1.1 *Algorithm 1: Using uncertainty to select the mask*

We provide a template algorithm utilising uncertainty quantification for the masking strategy in algorithm 1, given a sequence $x$ from the training set, model $G$ as our model to predict nucleotides at each position, $u(x_i)$ as our uncertainty quantification method being applied to a nucleotide at position $i$ in the sequence, and $m$ as our masking percentage.

Within this algorithm, we use model $G$ to generate predictions of each class at each nucleotide. We then iterate through each nucleotide, calculate the uncertainty score using $u(x_i)$, and record the lowest uncertainty score. After this, we sort through all positions in $x$, and select the top $m$ nucleotides from the sorted array. Lastly, we apply masking at these positions to get the final masked sequence.

---

**Algorithm 1** Masking Sequence based on Uncertainty

---

1: **Input:** Sequence $x$ with $L$ positions, masking percentage $m$, genomic foundation model $G$
2: **for** each position $i$ in sequence $x$ **do**
3:     Use model $G$ to predict nucleotide at position $x_i$
4:     Calculate uncertainty score $u(x_i)$ using softmax entropy or temperature scaling
5: **end for**
6: Sort all positions $i$ by uncertainty score $u(x_i)$ in descending order
7: Select the top $m$ positions with the highest uncertainty scores
8: **Output:** Masked sequence $x_m$

---

### 3.1.2 *Algorithm 2: Using uncertainty to select the input class.*

We provide a second template algorithm, algorithm 2, which utilises the uncertainty predictions obtained in algorithm 1. This algorithm demonstrates our approach for using these predictions to select the nucleotide bases and augmenting our sequence with new data. Let the sequence $x$ have $i$ positions, and $M$ represents the set of masked tokens within the sequence. $x_m$ represents the masked sequence created in algorithm 1. Furthermore, let class $c$ have $j$ nucleotides, and $C$ represents the set of nucleotides that it can choose from. For each position $i \in M$, and each class $j \in C$, we compute the uncertainty score $u(c_j \mid x_i)$. $P_M$ represents the predicted masked nucleotides, where

the class with the lowest uncertainty score is selected for each masked nucleotide. $u(c_j \mid x_i)$ is the uncertainty score assigned to class $c_j$ at position $x_i$, and $\hat{c}(x)$ is the class with the lowest uncertainty score, or the true structural label if it did not meet our cut-off value, $y$.

We take the masked predictions of bases $(P_M)$ and inject them directly into the sequence, thus creating the augmented sequences. By following this approach, we systematically augment each sequence until the final set of augmented sequences is generated.

---

**Algorithm 2** Uncertainty-based Sequence Augmentation

---

1: **Input:** Masked Sequence $x_m$ with $L$ positions, set of classes $C$, threshold $y$ for uncertainty score
2: **Output:** Augmented sequence $P_M$
3: $M \leftarrow$ set of masked positions in sequence $x$
4: $P_M \leftarrow x_m$
5: **for** each masked position $i \in M$ **do**
6:     **for** each class $c_j \in C$ **do**
7:         Compute uncertainty score $u(c_j \mid x_i)$
8:     **end for**
9:     $\hat{c}(x_i) \leftarrow \arg\min u(c \mid x_i)$
10:     **if** $\hat{c}(x_i) \leq y$ **then**
11:         $P_M(x_i) \leftarrow \hat{c}(x_i)$
12:     **else**
13:         $P_M(x_i) \leftarrow$ true structural label
14:     **end if**
15: **end for**
16: **Return** $P_M$

---

### 3.2 Uncertainty Quantification Methods

In this section we outline the uncertainty quantification methods used within this work, and provide insight into how this may affect the data augmentation results.

- **Softmax Entropy** The softmax function transforms the raw outputs of the model into a probability distribution over the classes. Softmax alone is known to be prone to overestimation and is unsuitable for uncertainty quantification (Holm et al., 2023), therefore we use the probabilities obtained through softmax to calculate the entropy value for each possible class.
- **Temperature Scaling** Temp Scaling learns a parameter (T or temp) for all classes. The idea is to "soften" the softmax with T > 1 (i.e., raise the entropy), and to "harden" the softmax with T < 0 (i.e., lower the entropy) (Guo et al., 2017). The exact parameter (T) is optimised with respect to Negative Log Likelihood (NLL) on the validation set.

## 4 Experimental Design

This section presents the framework used to empirically validate the proposed data augmentation methodology applied to several RNA foundation models extended to the pseudoknot-inclusive prediction problem. Additionally, we compare with current thermodynamic and dynamic programming approaches, although as these algorithms do not require training data, our augmentation method cannot be incorporated within their method. Consequently, they serve to verify the validity of our results. Two primary datasets are utilised within this experimental validation, bpRNA-PKinc and RNAStrAlign-PKinc, both of which have been curated from publicly available works, and further designed explicitly for the pseudoknot-inclusive prediction problem. Lastly, we provide a justification for the metrics employed for algorithm comparison.

Furthermore, we introduce a detailed analysis, designed to evaluate the uncertainty quantification methodologies incorporated within the augmentation strategy. This analysis is comprised of three parts: a demonstration of the UQ methodology's effect on the augmentation process, an investigation into the impact of hyper-parameters introduced by the methodology, and an ablation study to investigate the most effective areas of our UQ methodologies.

## 4.1 DATASETS

We propose two key datasets for benchmarking pseudoknot-inclusive secondary structure prediction algorithms. We outline the key properties of the training, validation, and testing splits of these datasets as detailed in table 1. These datasets were curated using CD-HIT-EST (Fu et al., 2012) with an 80% sequence similarity threshold, ensuring accuracy and reliability with respect to real RNA sequences, and further preventing data leakage from structurally similar sequences being included within training and testing datasets. A central challenge with pseudoknot-inclusive algorithms is the inherent class imbalance of pseudoknots within the secondary structures. To address this issue, we include two large datasets, exclusively containing pseudoknot-inclusive structures, thereby reducing the extent of the imbalance. Both datasets were partitioned into training, validation, and testing sets with an 80%, 10%, 10% split. Additionally, all GFMs were re-trained on each training set to remove the risk of data leakage.

- **bpRNA-PKinc (Danaee et al., 2018):** bpRNA consists of RNA sequences and secondary structures collected from seven databases, (Comparative RNA Web, tmRNA, tRNAdb, Signal Recognition Particle, RNaspe P, tRNAdb 2009 and RCSB Protein Data Bank) and annotated via the bpRNA tool.
- **RNAStrAlign-PKinc (Saman Booy et al., 2022):** The original dataset released by Saman Booy et al. (2022) contains 30, 451 sequence and structures from eight RNA families (16S, 5S, Group I Intron, RNaseP, SRP, telomerase, tmRNA, tRNA) post-filtering, with sequence lengths between 30 and 1851 nucleotides. Following filtering with CD-HIT-EST to remove redundant sequences with a similarity greater than >80%, we further removed all data without pseudoknots. The finalised dataset is presented in table 1.

Table 1: Dataset Overview: Training, Validation, and Testing Data

| Training Data | | | | |
|---|---|---|---|---|
| **Dataset** | **Seq. Count** | **Min Len** | **Max Len** | **Source** |
| bpRNA-PKinc | 1343 | 21 | 4216 | (Danaee et al., 2018) |
| RNAStrAlign-PKinc | 1152 | 163 | 1587 | (Saman Booy et al., 2022) |
| **Validation Data** | | | | |
| **Dataset** | **Seq. Count** | **Min Len** | **Max Len** | **Source** |
| bpRNA-PKinc | 169 | 31 | 4381 | (Danaee et al., 2018) |
| RNAStrAlign-PKinc | 144 | 209 | 1559 | (Saman Booy et al., 2022) |
| **Testing Data** | | | | |
| **Dataset** | **Seq. Count** | **Min Len** | **Max Len** | **Source** |
| bpRNA-PKinc | 167 | 28 | 3420 | (Danaee et al., 2018) |
| RNAStrAlign-PKinc | 233 | 144 | 1574 | (Saman Booy et al., 2022) |

## 4.2 GENOMIC FOUNDATION MODELS

In this section, we outline the algorithms employed in this study. We begin with IPKnot++ (Sato et al., 2021) and pKiss (Janssen & Giegerich, 2014). The latter of which is recognised as the leading thermodynamic method for pseudoknot prediction, while the former uses integer programming to navigate the search space. Both algorithms do not require training data, and thus are incompatible with our augmentation method. Whilst other deep learning approaches exist in current literature (e.g., Ufold (Fu et al., 2021)), they utilise a differing data-format known as base-pair sequence (BPSEQ). As our augmentation method requires the secondary structures to be in dot-bracket notation form, they were excluded from this analysis.

Next, we describe the three foundation models used to demonstrate the efficacy of our method. The first, OmniGenome-52M (Yang & Li, 2024), is an RNA-based architecture pre-trained on the OneKP dataset (Initiative, 2019), utilising a transformer encoder-only architecture with 52 million parameters. The second, HyenaDNA (Nguyen et al., 2024b), is a DNA-based model pre-trained on the Human Reference Genome dataset (Consortium, 2013), featuring a transformer decoder-only, sequence-to-sequence architecture with 47 million parameters. Lastly, SpliceBERT (Chen et al., 2024) is an RNA-based model pre-trained on RNA sequences from 72 vertebrates using MLM, employing a transformer encoder-only architecture with 19.7 million parameters.

These models were specifically selected due to their utilisation of Single Nucleotide Tokenisation (SNT). Whilst other foundation models for RNA and DNA are present in current literature, their applicability to this problem is hindered by the usage of K-mers or Byte Pair Encoding (BPE) tokenisation. These methods may aggregate multiple nucleotides into single tokens, which could misalign the sequence and the secondary structure. As we translate each nucleotide from the original sequence into its corresponding secondary structure token-by-token, nucleotides being merged together may cause a de-sync between the structure produced and the original sequence length.

### 4.3 METRICS

In this section we outline the metrics utilised to assess the performance of the secondary structure prediction algorithms.

- **F1-Score** - F1-Score represents the harmonic mean of the precision and recall. Precision measures the proportion of correct positive predictions out of all positive predictions made, and recall measures the proportion of correctly predicted positive instances out of all actual positive instances. F1 measures the number of correct tokens for each target structure.

- **Matthew's Correlation Coefficient** - Matthew's Correlation Coefficient (MCC) averages the model scores across all possible classes, thereby showing model performance even when there is a large class imbalance. This, combined with F1-Score, is crucial for demonstrating model performance on the pseudoknot-inclusive problem, due to the significant class imbalance in the structural contents (Chicco & Jurman, 2020).

- **Accuracy** - Accuracy measures the percentage of correct predictions by the algorithm, in this case, the number of correctly predicted nucleotides within the structure. Accuracy can be misleading due to over-emphasising the class imbalanced data, thus F1 and MCC are included for comparison.

### 4.4 UNCERTAINTY QUANTIFICATION

We aim to investigate the point of uncertainty at which algorithm performance improves. Our hypothesis posits that identifying the position of nucleotides that exhibits high uncertainty, and subsequently masking those nucleotides during data augmentation, the algorithm can improve its effectiveness in previously weak areas, thereby improving overall performance. To test this hypothesis, we re-apply the augmentation methodology on both bpRNA and RNAStrAlign datasets and re-train the three algorithms on each augmented dataset.

This ablation study consists of two primary aspects. First, we aim to gain insight into the relationship between augmented and non-augmented sequence similarity, and how this affects algorithm performance. Intuitively, we anticipate that high similarity may not improve model performance, as repeated data could lead to overfitting. Conversely, a high mutation rate may cause hallucinations. We hypothesise that the optimal value lies between 10% to 50%. To evaluate our hypothesis, we implement the augmentation strategy with a 10% variation at each interval, ranging from 10% to 50%.

Secondly, we aim to examine the relationship between the number of augmentations and model performance. It has previously been proven that increasing the proportion of augmented data does not necessarily correlate with improved performance, it may instead cause the model to over-fit on the oversampled minority classes (Shorten & Khoshgoftaar, 2019). However, the model must receive sufficient augmented data to effectively capture the class imbalance, and discern nucleotides relationships within the RNA sequence, and their relationship with the secondary structure. Therefore, we conduct an ablation study on the ratio of augmentations to non-augmented data, to gain insight into the efficacy of our method across both small and large amounts of augmented data.

## 5 RESULTS

This section presents the results of our data augmentation methodology, and further assess the results of our ablation experiments evaluating the newly introduced hyper-parameters.

## 5.1 Performance without augmentation

Given the major class imbalance within both the training and testing data, we find it prudent to analyse model performance on zero-shot data, where no pseudoknot-inclusive data is included during fine-tuning. We further demonstrate model performance without data augmentation in table 2, as to allow for a comparison with the data augmentation methods.

The zero-shot performance is outlined in fig. 2. It is evident that without including pseudoknot-inclusive data, the models struggle to generalise and predict pseudoknots effectively. The three GFMs have been trained on a diverse array of datasets, (DNABERT on DNA data, SpliceBERT on RNA splicings, and OmniGenome on RNA data from 1KP) most of which is pseudoknot-free. Consequently it is unremarkable that the models exhibit difficulty in adapting to the class imbalanced data.

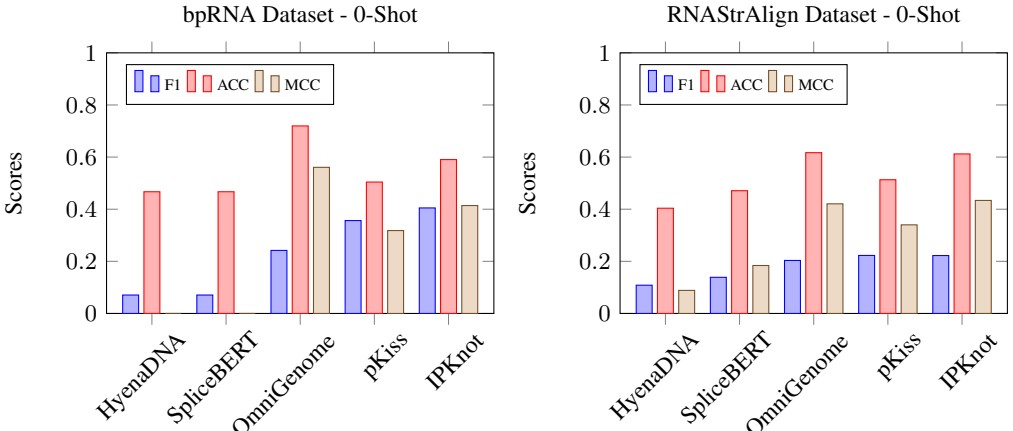

Figure 2: 0-shot results for bpRNA (left) and RNAStrAlign (right). We train all three foundation models on data with no pseudoknots, and test on our pseudoknot-inclusive testing datasets.

The performance of the fine-tuned GFMs on our pseudoknot-inclusive datasets indicates that deep learning models can generalise effectively to pseudoknot-inclusive data. However, the overall performance of our GFMs is significantly lower on bpRNA, which is skewed towards a shorter distribution of sequences. This is in contrast with the thermodynamic (pKiss) and integer programming (IPKnot) approaches, which notably see a significant performance drop with the longer sequence length distribution in RNAStrAlign. Despite this, it is evident that OmniGenome achieves state-of-the-art performance across both testing sets, thereby demonstrating the effectiveness of GFMs when appropriately adapted to address the pseudoknot-inclusive prediction problem.

## 5.2 Enhanced Comparison Models

We observe that OmniGenome-52M with data augmentation demonstrates superior performance across both datasets, and with all three uncertainty quantification strategies. With data augmentation, it emerges as the state-of-the-art method, greatly surpassing the previous thermodynamic and integer programming methods of pKiss and IPKnot. We further compare augmented model performance with non-augmented performance on the imbalanced classes. As detailed in fig. 3, augmented models see a significant improvement on the under-represented classes over non-augmented models.

Our findings indicates the application of data augmentation, combined with an uncertainty estimator, yields improvements across both datasets, and for all augmented models. Notably, HyenaDNA, the only DNA-based GFM, demonstrates the least improvement among the three GFMs, whereas SpliceBERT, the model with the smallest parameter size, finds the greatest improvement across both datasets. We observe a significant enhancement in performance for all models within the bpRNA dataset. However, for the RNAStrAlign dataset, the transition from no augmentation to optimal augmentation yields negligable improvement, and only a modest improvement for OmniGenome. This discrepancy is likely a result of the higher consistency of RNA sequences within the RNAS-trAlign dataset, as bpRNA features a much greater diversity of sequence length and pseudoknot

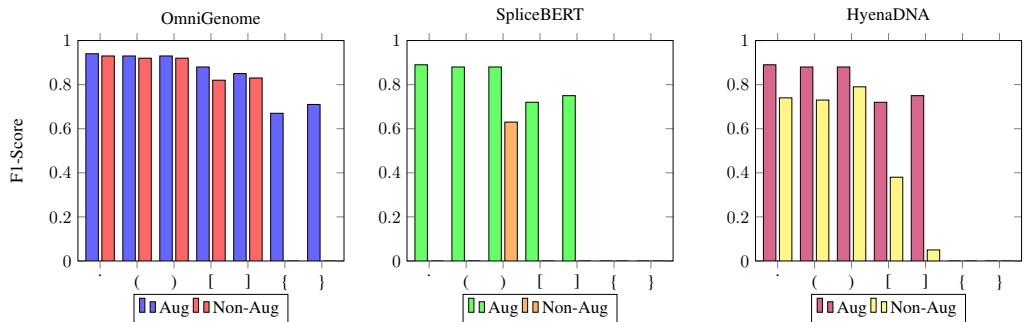

Figure 3: Comparison of Augmented and Non-Augmented macro-F1 class results for OmniGenome, SpliceBERT, and HyenaDNA on the bpRNA dataset

Table 2: Best F1-Score of models on testing datasets with uncertainty estimation methods

| Model | Dataset | TS | SME | Rand | No Aug |
|-------|---------|-----|------|------|--------|
| **OmniGenome** | bpRNA | 63.10 | 66.55 | 44.23 | 48.83 |
| | RNAStrAlign | 87.32 | 89.73 | 86.43 | 87.28 |
| **HyenaDNA** | bpRNA | 40.39 | 41.35 | 30.63 | 30.32 |
| | RNAStrAlign | 84.88 | 85.02 | 48.22 | 84.62 |
| **SpliceBERT** | bpRNA | 46.96 | 47.21 | 37.66 | 07.00 |
| | RNAStrAlign | 60.19 | 83.11 | 47.42 | 67.16 |

types. Consequently, the original RNAStrAlign dataset provides sufficient structural and sequential information for the model to develop an understanding of the underlying structural patterns, reducing the effectiveness of our data augmentation method.

table 2 consistently demonstrates the importance of uncertainty quantification method over random mutation, as it leads to improved better model performance across all datasets. Moreover, softmax entropy (SME) consistently outperforms Temperature Scaling (TS) across nearly all models and datasets, although the disparity between the two methods is minimal in most cases. Temperature scaling is typically superior to SME at measuring model uncertainty and mitigating out-of-distribution overestimation (Guo et al., 2017). However, by incorporating all pseudoknot types in both training and testing datasets, we minimise out-of-distribution data. This allows SME to more accurately capture model uncertainty compared to temperature scaling.

## 5.3 ABLATION STUDY ANALYSIS

### 5.3.1 SEQUENCE SIMILARITY

The hyper-parameter introduced in this study, the percentage of augmented sequences, demonstrates a strong dependence on the dataset utilised. Our ablation study reveals that the percentage of nucleotides masked directly influences model performance, although this relationship is not directional. An increase in model masking may cause a positive or negative effect, due to the relationship being dependent on the original dataset utilised. Our initial hypothesis posited that an increased percentage of nucleotides may result in an increased performance for similar sequences through introducing diversity amongst the results. Conversely, a low percentage of nucleotides may result a high similarity between sequences, and could cause overfitting. This phenomena is directly observed when using random masking, however we mitigate this with the introduction of uncertainty quantification. This stems from the uncertainty quantification method providing additional insight into areas with low proportions within the dataset, thereby enabling prediction re-calibration.

Table 3: F1-Score of models on bpRNA with various SLM augmentation percentages

| Model | 10% | 20% | 30% | 40% | 50% |
|---|---|---|---|---|---|
| **OmniGenome** | 65.43 | 65.78 | 66.55 | 65.11 | 65.73 |
| **HyenaDNA** | 39.86 | 39.66 | 40.39 | 41.07 | 41.35 |
| **SpliceBERT** | 07.00 | 46.51 | 46.13 | 47.21 | 45.85. |

### 5.3.2 NUMBER OF AUGMENTED SEQUENCES

Lastly, we performed an investigation into our second ablation study examining the impact of the number of augmented sequences on model performance. It has been documented in computer vision tasks that an increase in augmented data improves model performance, until a point of diminishing returns is reached (Xie et al., 2019). We present ratios of 1:0, 1:2, 1:5, 1:8, 1:10 and 1:12, with our ablation study indicating that 1:10 is the point at which the augmentation method is most effective. A noteworthy observation is that the model utilised drastically influences the number of data augmentations needed before performance is significantly affected. For instance, HyenaDNA only requires

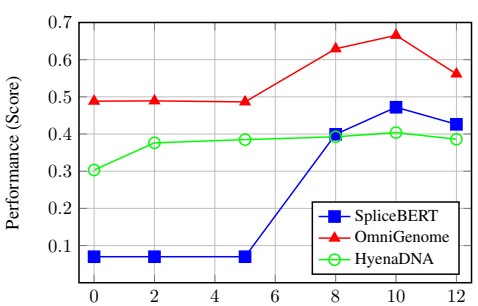

Figure 4: Performance vs Augmentation Level for Different Models on bpRNA Dataset

the minimum augmentation, whereas SpliceBERT and OmniGenome necessitate at least 8 augmentations per sequence to achieve notably improvements. Additionally, we observe that increasing the number of augmentations generally positively affects model performance, although with gradually reducing returns.

## 6 CONCLUSION

We propose Sequence Location Masking (SLM) in this work to address the challenges of pseudoknot-inclusive secondary structure prediction (SSP). Our approach demonstrates state-of-the-art performance on both bpRNA and RNAStrAlign test datasets, outlining the effectiveness of our data augmentation technique across a diverse range of sequence lengths and pseudoknot types. Additionally, we conducted an ablation study to investigate the hyper-parameters introduced, analysed their impact on model performance, and identified the critical parameters that improve model performance. Future work should aim to expand the applicability of SLM by exploring its integration with other types of deep learning models outside of GFMs. SLM should further be investigated in the context of larger and more diverse genomic tasks to evaluate its robustness and adaptability.

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

# A APPENDIX

## A.1 TABLE OF BEST RESULTS

We further provide a complete table of the best model performance with the three key metrics outlined in section 4.

Table 4: Best Model Performance on bpRNA and RNAStrAlign Datasets

| Model | bpRNA Dataset | | | RNAStrAlign Dataset | | |
|---|---|---|---|---|---|---|
| | F1 | ACC | MCC | F1 | ACC | MCC |
| OmniGenome | 63.10 | 93.50 | 90.13 | 89.73 | 98.02 | 97.04 |
| SpliceBERT | 47.21 | 88.54 | 82.05 | 83.11 | 96.66 | 95.00 |
| HyenaDNA | 41.35 | 79.62 | 68.81 | 84.64 | 94.37 | 91.57 |
| pKiss | 35.64 | 50.42 | 31.80 | 22.26 | 51.31 | 34.00 |
| IPKnot | 39.69 | 59.33 | 39.45 | 22.22 | 61.21 | 43.37 |

