# OpenReview forum: "Data Augmentation via Genomic Foundation Models for Pseudoknot-Inclusive RNA Secondary Structure Prediction"
_ICLR.cc/2025/Conference — ICLR 2025 Conference Withdrawn Submission_

### Official Review · Reviewer_EnjT · 2024-10-21

**Soundness:** 1
**Presentation:** 1
**Contribution:** 2
**Rating:** 3
**Confidence:** 5

**Summary:**

The work proposes a data augmentation strategy for pseudoknot-inlcusive RNA secondary structure prediction based on uncertainty quantification. Using the proposed augmentation strategy, the authors claim state-of-the-art performance on two new datasets. In addition, the authors analyze their method in an ablation study.

**Strengths:**

- The paper addresses an important problem of computational biology, namely RNA secondary structure prediction with pseudoknots.
- The paper is well written in terms of language.
- The proposed augmentation technique seems to be beneficial for the specific models used in the study.

**Weaknesses:**

Major issues:

- The authors claim state-of-the-art performance, however, while the RNAStrAlign and BpRNA-1m datasets have been widely used in previous work, the authors unnecessarily introduce a new training/validation/test split which makes the results incomparable to previous work. For example in [1], a PK-Test set, split from the BpRNA (and Rfam) set, is used with a proposed model showing strong performance (F1-score > 70). Furthermore, there are existing testsets, TS0-3 and TS-hard, introduced in [2] and [3], that are regularly used for benchmarking of RNA secondary structure prediction. Why do the authors have to introduce a new dataset? Why do the authors not simply use the PK-structures of the existing sets to be more comparable?
- A comparison to sota methods is completely lacking. IPKnot and PKiss are currently *NOT* considered state-of-the-art. In addition PKiss can predict only a subset of PKs. This is not mentioned in the paper.
- The timeliness of the work is questionable. Since the augmentation technique only considers secondary structures represented in dot-bracket notation, new state-of-the-art deep learning methods cannot benefit from the method. In addition, the representation of RNA secondary structures as adjacency matrices is one of the key advantages of recent deep learning based methods compared to more traditional, dynamic programming approaches. This limitation is not discussed at all in the paper.
- The authors claim to introduce two new pseudoknot-inclusive datasets while preventing “data leakage from structurally similar sequences”. However, the data is split using CD-Hit, which only considers sequence similarity. Several recent publications raised concerns about splitting RNA data based on sequence similarity only, which typically results in homologous training and test data (see e.g. [4, 5, 6]). Since RNA is often more conserved in structure, applying structural measures is crucial during data splitting.
- The authors do not include deep learning based approaches in their experiments because “they utilise a differing data-format … (BPSEQ)”. This is not generally true, for example [7] is based on string representations in dot-bracket format.
- The proposed method only works for Single Nucleotide Tokenization Methods. This clearly is a limitation of the proposed method but it is sold as if k-mer or BPE tokenization is a limitation of other foundation models. While different tokenization methods are still discussed in the literature for genomic foundation models, this limitation of the proposed approach is not discussed in the paper. As a side note, [7] uses single nucleotide tokenization.
- Generally, the work doesn’t show much novelty, is very specific, and I don't think that it does align very much with the ICLR audience.

Minor aspects:

- [Line 055/056] “Yet these benchmarks do not consider the pseudoknot…”
This is not true. At least the cited RNABench contains tasks including pseudoknot predictions. Moreover, RNABench includes tasks that consider even base multiplets, which cannot be represented in dot-bracket notation.

- [Lines 067-070] “[...] NP-Hard, thus dynamic programming-based methods require polynomial time [...]”
This sentence contains a contradiction unless P = NP.

- [Line 079] “[...] researchers have increasingly turned to data augmentation [...]”
The cited work in [8] to support this claim does not use data augmentations.

- The work ignores the fact that the BpRNA dataset reports pseudoknots up to a page number of 7 (although only a very small fraction of PKs falls into this category), while the authors only consider a page number of 4. However, rare PKs would be a very interesting example for the work, because of even stronger class imbalance.

- I might have missed it but the abbreviation UQ (Line 268) is not introduced.


[1] Gong, T., Ju, F., & Bu, D. (2024). Accurate prediction of RNA secondary structure including pseudoknots through solving minimum-cost flow with learned potentials. Communications Biology, 7(1), 297.

[2] Singh, J., Hanson, J., Paliwal, K., & Zhou, Y. (2019). RNA secondary structure prediction using an ensemble of two-dimensional deep neural networks and transfer learning. Nature communications, 10(1), 5407.

[3] Singh, J., Paliwal, K., Zhang, T., Singh, J., Litfin, T., & Zhou, Y. (2021). Improved RNA secondary structure and tertiary base-pairing prediction using evolutionary profile, mutational coupling and two-dimensional transfer learning. Bioinformatics, 37(17), 2589-2600.

[4] Flamm, C., Wielach, J., Wolfinger, M. T., Badelt, S., Lorenz, R., & Hofacker, I. L. (2022). Caveats to deep learning approaches to RNA secondary structure prediction. Frontiers in Bioinformatics, 2, 835422.

[5] Szikszai, M., Wise, M., Datta, A., Ward, M., & Mathews, D. H. (2022). Deep learning models for RNA secondary structure prediction (probably) do not generalize across families. Bioinformatics, 38(16), 3892-3899.

[6] Qiu, X. (2023). Sequence similarity governs generalizability of de novo deep learning models for RNA secondary structure prediction. PLOS Computational Biology, 19(4), e1011047.

[7] Franke, J., Runge, F., & Hutter, F. (2022). Probabilistic transformer: Modelling ambiguities and distributions for rna folding and molecule design. Advances in Neural Information Processing Systems, 35, 26856-26873.

[8] Runge, F., Stoll, D., Falkner, S., & Hutter, F. Learning to Design RNA. In International Conference on Learning Representations.

**Questions:**

(see also weaknesses)

- Why did the authors not include commonly used datasets from the literature? An evaluation on existing sets would make the work much more comparable.
- How does the method work for more recent DL methods like the Probabilistic Transformer mentioned in [7].
- What are the exact changes made to the GFMs in order to apply them to RNA secondary structure prediction? A discussion at least in the appendix would be valuable.

[7] Franke, J., Runge, F., & Hutter, F. (2022). Probabilistic transformer: Modelling ambiguities and distributions for rna folding and molecule design. Advances in Neural Information Processing Systems, 35, 26856-26873.

---

### Official Review · Reviewer_f4NG · 2024-10-28

**Soundness:** 1
**Presentation:** 3
**Contribution:** 1
**Rating:** 3
**Confidence:** 4

**Summary:**

This paper presents Sequence Location Masking (SLM), a data augmentation technique for RNA secondary structure prediction with pseudoknots. The authors combine masked language modeling with uncertainty quantification to generate synthetic data, aiming to address data scarcity and class imbalance. They evaluate their method by extending three genomic foundation models and testing on two datasets.

**Strengths:**

- The paper is clear and well written.
- The paper addresses an important challenge in computational biology.

**Weaknesses:**

- The paper compares RNA secondary structure prediction approaches based on dot-bracket but misses the Probabilistic Transformer [1] as a comparison.
- The authors argue that they compare not to other methods since their “method requires the secondary structures to be in dot-bracket notation form” missing that an adjacency matrix can be converted in dot-bracket notation. Therefore, the paper lacks important and well-performing related work such as SPOT RNA[2], SPOT RNA2[3], or the recently published RNAformer [4].
- The authors do not adequately consider the homology between training and test data, see [5].
- The paper lacks an evaluation of the proposed method on high-quality data e.g. from the protein database (PDB). It constrains over 6000 RNA structures.
- No code submission to replicate the results.
___

[1] Franke, J., Runge, F., & Hutter, F.: Probabilistic Transformer: Modelling ambiguities and distributions for RNA folding and molecule design. Adv. Neural Inf. Process. Syst. 35, 1-12 (2022).

[2] Singh, J., Hanson, J., Paliwal, K., & Zhou, Y.: RNA secondary structure prediction using an ensemble of two-dimensional deep neural networks and transfer learning. Nat. Commun. 10, 5407 (2019).

[3] Singh, J., Paliwal, K., Zhang, T., Singh, J., Litfin, T., & Zhou, Y.: Improved RNA secondary structure and tertiary base-pairing prediction using evolutionary profile, mutational coupling and two-dimensional transfer learning. Bioinformatics 37, 2589-2600 (2021).

[4] Franke, J., Runge, F., & Hutter, F.: Scalable Deep Learning for RNA Secondary Structure Prediction. In: ICML Workshop Comput. Biol., Honolulu, Hawaii, USA (2023).

[5] Flamm, C., Wielach, J., Wolfinger, M. T., Badelt, S., Lorenz, R., & Hofacker, I. L.: Caveats to deep learning approaches to RNA secondary structure prediction. Front. Bioinform. 2, 835422 (2022).

**Questions:**

- How would the augmentation perform in combination with SOTA RNA secondary structure prediction algorithms?
- Could you please add your code for reproducibility?

---

### Official Review · Reviewer_JSLt · 2024-11-02

**Soundness:** 3
**Presentation:** 2
**Contribution:** 3
**Rating:** 5
**Confidence:** 4

**Summary:**

The paper presents an innovative approach to address RNA secondary structure prediction (SSP) with pseudoknot-inclusive features. By utilizing a novel data augmentation technique through masked language modeling (MLM) and uncertainty quantification, the authors enhance genomic foundation models (GFMs) for improved SSP performance.

**Strengths:**

1. The approach leverages advances in GFMs to tackle RNA SSP, addressing the specific challenge of pseudoknot prediction. The data augmentation method appears thoughtfully designed, targeting key limitations in existing SSP datasets.
2. The reported improvements in F1-score and MCC across both datasets illustrate the method’s impact on imbalanced class representation, particularly pseudoknot-inclusive structures.

**Weaknesses:**

1. The description of the methodology is unclear in certain parts. The specific steps for implementing Sequence Location Masking are difficult to understand at a high level, which creates some confusion about how the method works. Including a diagram could help clarify this process.

2. The paper does not compare its approach with other RNA secondary structure prediction methods, such as UFold or SPOT-RNA, which limits the assessment of the method’s relative performance.

3. There are no experiments conducted on the ArchiveII dataset.

4. In Figure 3, it is unclear how the F1-score is calculated separately for each type of base pair.

5. In Table 2, the "random mutation" method is not well-defined, and it is unclear why SpliceBERT performs so poorly with data augmentation.

**Questions:**

Please see Weaknesses.

---

### Note · Authors · 2024-11-29

**Comment:**

We sincerely the reviewers for their valuable and detailed feedback of our manuscript. After careful consideration, we have decided to withdraw the manuscript to allow us to thoroughly address the feedback and revise the work accordingly.

We greatly appreciate the time and effort spent by the reviewers in evaluating our submission. Their feedback has provided us with a clear path to improve the quality and impact of our work. Thank you for your efforts in making this research area better through your thoughtful reviews.

**Withdrawal Confirmation:**

I have read and agree with the venue's withdrawal policy on behalf of myself and my co-authors.